# Insulin/Snail1 axis ameliorates fatty liver disease by epigenetically suppressing lipogenesis

Yan Liu[1], Lin Jiang[1], Chengxin Sun[1], Nicole Ireland[1], Yatrik M. Shah[1,2], Yong Liu[3] & Liangyou Rui [1,2]

Insulin stimulates lipogenesis but insulin resistance is also associated with increased hepatic lipogenesis in obesity. However, the underlying mechanism remains poorly characterized. Here, we show a noncanonical insulin-Snail1 pathway that suppresses lipogenesis. Insulin robustly upregulates zinc-finger protein Snail1 in a PI 3-kinase-dependent manner. In obesity, the hepatic insulin-Snail1 cascade is impaired due to insulin resistance. Hepatocyte-specific deletion of *Snail1* enhances insulin-stimulated lipogenesis in hepatocytes, exacerbates dietary NAFLD in mice, and attenuates NAFLD-associated insulin resistance. Liver-specific over-expression of Snail1 has the opposite effect. Mechanistically, Snail1 binds to the fatty acid synthase promoter and recruits HDAC1/2 to induce deacetylation of H3K9 and H3K27, thereby repressing fatty acid synthase promoter activity. Our data suggest that insulin pathways bifurcate into canonical (lipogenic) and noncanonical (anti-lipogenesis by Snail1) two arms. The noncanonical arm counterbalances the canonical arm through Snail1-elicited epigenetic suppression of lipogenic genes. Impairment in the insulin-Snail1 arm may contribute to NAFLD in obesity.

[1] Department of Molecular & Integrative Physiology, University of Michigan Medical School, Ann Arbor, MI 48109, USA. [2] Department of Internal Medicine, University of Michigan Medical School, Ann Arbor, MI 48109, USA. [3] College of Life Sciences, the Institute for Advanced Studies, Wuhan University, Wuhan 430072, China. Correspondence and requests for materials should be addressed to L.R. (email: ruily@umich.edu)

Prevalence of nonalcoholic fatty liver disease (NAFLD) increases in an alarming pace due to the obesity epidemic[1]. The outcomes of NAFLD are ominous, including insulin resistance, type 2 diabetes, dyslipidemia, cardiovascular disease, liver fibrosis, cirrhosis, and/or hepatocellular carcinoma[1–3]. Liver lipid levels are determined by an interplay between de novo lipogenesis, lipid uptake, fatty acid β oxidation, and very low-density lipoprotein (VLDL) secretion. Notably, hepatic lipogenesis increases in NAFLD[4,5], and genetic disruption of the hepatic lipogenic program prevents NAFLD[6–8]. Thus, inhibiting hepatic lipogenesis likely provides a therapeutic strategy for combatting NAFLD.

Liver lipogenesis is regulated predominantly by metabolic hormone insulin. Insulin stimulates the canonical lipogenic pathway, including activation of lipogenic transcription factors liver X receptor (Lxr), Srebp-1c, and upstream stimulatory factor-1 (Usf-1)[5,9]. These nuclear proteins activate expression of lipogenic enzymes ATP citrate lyase (Acl), acetyl coenzyme A carboxylase 1 (Acc1), and/or fatty acid synthase (Fasn)[5]. Paradoxically, insulin resistance is associated with increased hepatic lipogenesis in obesity, contributing to NAFLD[10]. However, the underlying mechanism remains poorly understood.

We recently reported that insulin upregulates adipose Snail1 that in turn suppresses expression of adipose triacylglycerol lipase (ATGL) and ATGL-mediated lipolysis[11]. Snail1 is a transcriptional repressor, and has been known to induce epithelial-to-mesenchymal transition (EMT) during development or in cancer metastasis[12–14]. Snail1 has been well documented to epigenetically suppress expression of E-cadherin and claudin, thus promoting EMT[15–18]. Mechanistically, Snail1 binds via its N-terminal SNAG domain to several epigenetic enzymes, including histone deacetylases (HDACs) and histone methyltransferases, and recruit them to target promoters where these enzymes catalyze repressive histone modifications[14,15]. Notably, we found two reports that describe the potential action of hepatic Snail1 in liver injury and regeneration[19,20]; however, the metabolic function of hepatic Snail1 has not been explored. In this study, we provide proof of concept evidence showing that hepatic Snail1 is an unrecognized suppressor of de novo lipogenesis. It epigenetically represses expression of lipogenic enzymes. We further demonstrate that insulin robustly upregulates Snail1 which defines the noncanonical anti-lipogenic pathway. Thus, this work unravels a bifurcation of insulin signaling into the canonical lipogenic and the noncanonical anti-lipogenic arms.

## Results

**Insulin upregulates hepatic Snail1 via PI 3-kinase pathway.** Given that insulin stimulates Snail1 expression in adipocytes[11], we postulated that insulin might similarly upregulate Snail1 in hepatocytes. Indeed, insulin markedly increased Snail1 mRNA levels in both mouse primary hepatocytes and human HepG2 hepatocytes (Supplementary Fig. 1a, b), and substantially increased Snail1 protein levels in HepG2 hepatocytes (Fig. 1a). Likewise, insulin markedly increased the mRNA and protein levels of hepatic Snail1 in C57BL/6 mice (Supplementary Fig. 1c, d). Consistently, liver Snail1 levels were lower in the fasted state (low plasma insulin levels) than in the fed state (Supplementary Fig. 1e). To identify pathways responsible for upregulation of hepatic Snail1, we pretreated HepG2 hepatocytes with PI 3-kinase (wortmannin) or Akt (MK2066) inhibitors. Inhibition of either PI 3-kinase or Akt blocked upregulation of Snail1 by insulin (Fig. 1a and Supplementary Fig. 1f). Notably, Akt2 was reported to mediate TFGβ1-induced upregulation of Snail1[21]. These data suggest that the PI 3-kinase/Akt pathway is required for insulin to upregulate hepatic Snail1.

Given that GSK3β induces ubiquitination and degradation of Snail1[22,23], we speculated that insulin might suppress proteasome-mediated degradation of Snail1 via GSK3β. Insulin stimulated phosphorylation and inactivation of GSK3β (Fig. 1a), confirming that insulin inhibits GSK3β via the PI 3-kinase/Akt pathway[24]. First, we assessed Snail1 stability in HepG2 hepatocytes using protein synthesis inhibitor cycloheximide. Insulin profoundly inhibited degradation of Snail1 (Fig. 1b). Half-life of Snail1 was prolonged from 0.8 h in PBS-treated cells to 2.4 h in insulin-stimulated cells (Fig. 1c). Second, we measured ubiquitination of Snail1. We did not detect ubiquitinated Snail1 in DMSO-treated HepG2 cells (Fig. 1d), presumably due to rapid degradation. Hence, we blocked degradation by pretreating cells with proteasome inhibitor MG132. We detected robust ubiquitination of Snail1; importantly, insulin dramatically decreased the levels of ubiquitinated Snail1 (Fig. 1d).

We asked if the insulin/Snail1 axis is impaired in obesity, owing to insulin resistance. C57BL/6 mice were fed a high fat diet (HFD) for 6 weeks to induce obesity. Insulin treatment substantially increased hepatic Snail1 levels in chow-fed mice but not in HFD-fed mice (Fig. 1e and Supplementary Fig. 1g). To validate these findings in vitro, we pretreated HepG2 hepatocytes with palmitic acid to model metabolic environments in obesity. Palmitic acid pretreatment abrogated the ability of insulin to upregulate Snail1 (Fig. 1e and Supplementary Fig. 1h, i). Thus, the hepatic insulin/Snail1 axis is impaired in obesity.

**Insulin/Snail1 axis suppresses hepatic lipogenesis.** We sought to delineate the role of the insulin/Snail1 axis in hepatic lipogenesis. To disrupt this axis, we generated inducible, hepatocyte-specific Snail1 knockout (Snail1^{Δhep}) mice by crossing Snail1^{flox/flox} mice with albumin-CreER^{T2} drivers[25,26]. Snail1^{flox/flox};CreER^{T2} mice were treated with tamoxifen to obtain Snail1^{Δhep} mice. Snail1^{flox/flox} littermates were treated similarly with tamoxifen as control. Snail1 was disrupted in the livers but not white adipose tissue (WAT) of Snail1^{Δhep} mice (Supplementary Fig. 2a). We assessed lipogenesis in primary hepatocytes isolated from Snail1^{Δhep} and Snail1^{flox/flox} mice. Deletion of Snail1 substantially increased the ability of insulin to stimulate lipogenesis (Fig. 2a), and caused an upward shift in insulin dose response curves (Supplementary Fig. 2b). To corroborate these findings, we measured expression of key lipogenic enzymes Fasn, Acc1, and Acl. Snail1 deficiency markedly enhanced the ability of insulin to increase both mRNA (Fig. 2b) and protein (Fig. 2c) levels of these enzymes. Given upregulating Snail1 by insulin, these results unveil that the insulin/Snail1 axis inhibits lipogenesis in a feedforward manner.

To further confirm the antagonistic effect of the insulin/Snail1 axis on lipogenesis, we overexpressed Snail1 in primary hepatocytes using Snail1 adenoviral vectors. Green fluorescent protein (GFP) vectors were used as control. Overexpression of Snail1 completely blocked insulin-stimulated lipogenesis; notably, it also decreased baseline lipogenesis (Fig. 2d). In agreement with these results, Snail1 blocked insulin-stimulated expression of Fasn, Acc1, and Acl (Fig. 2e, f). Notably, Snail1 did not inhibit insulin-stimulated phosphorylation of Akt (Supplementary Fig. 2c), further supporting the notion that the insulin/Snail1 axis suppresses lipogenesis via a feedforward circuit. Snail1 did not alter the ability of insulin to suppress gluconeogenesis in primary hepatocytes (Supplementary Fig. 2d), suggesting that the insulin/Snail1 axis specifically regulates lipid but not glucose metabolism in hepatocytes. In light of these findings, we propose that insulin signaling bifurcates into the canonical (lipogenic) and the noncanonical Snail1 (anti-lipogenic) arms.

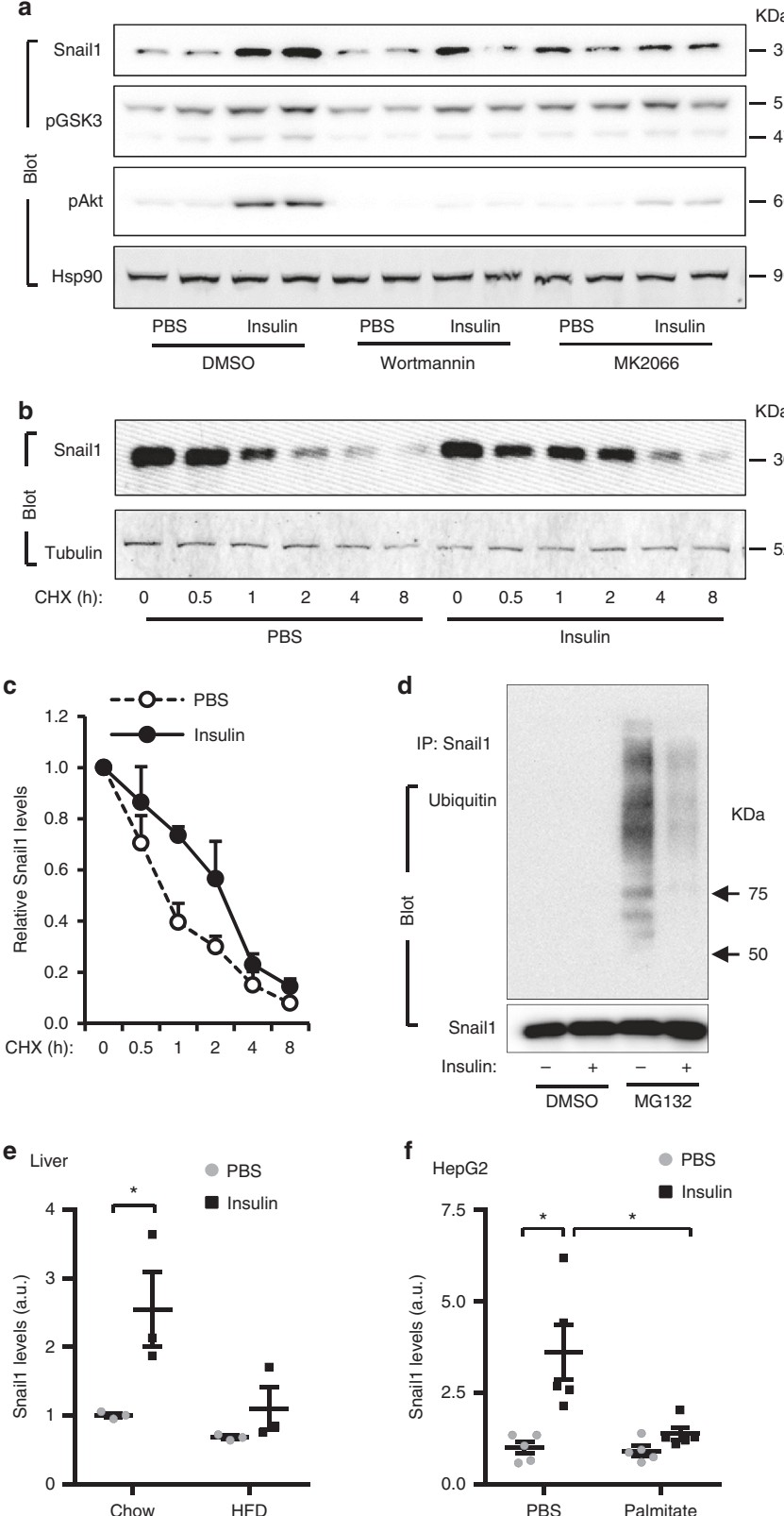

**Hepatocyte-specific deletion of *Snail1* promotes NAFLD**. We set out to explore the function of the noncanonical Snail1 arm in vivo by characterizing *Snail1^{Δhep}* mice. *Snail1^{Δhep}* mice were grossly normal. Body weight and plasma triacylglycerol (TAG) and nonesterified fatty acids (NEFA) were comparable between *Snail1^{Δhep}* and *Snail1^{flox/flox}* littermates (Supplementary Fig. 3a, b). Hepatocyte lipid droplets were larger and more abundant in *Snail1^{Δhep}* mice relative to *Snail1^{flox/flox}* littermates, as assessed by staining liver sections with neutral lipid dye Nile red (Fig. 3a). Consistently, liver TAG levels were significantly higher in

**Fig. 1** Insulin upregulates hepatic Snail1 via the PI 3-kinase pathway. **a** HepG2 cells were pretreated with wortmannin or MK2066, and then stimulated with insulin for 2 h. Cell extracts were immunoblotted with the indicated antibodies. Snail1 levels were normalized to Hsp90 levels ($n = 4$). **b**, **c** HepG2 cells were transfected with Snail1 plasmids and stimulated with insulin in the presence of cycloheximide. Cell extracts were immunoblotted with antibodies against Snail1 or α-tubulin. Snail1 levels (normalized to α-tubulin levels) were plotted against cycloheximide treatment durations ($n = 3$). **d** HepG2 cells were transduced with Snail1 adenoviral vectors and stimulated with insulin with or without MG132 (5 μM). Cell extracts were immunoprecipitated with antibody against Snail1 and immunoblotted with antibodies against ubiquitin or Snail1. **e** C57BL/6 males were fed a HFD or chow diet for 6 weeks, fasted overnight (14 h), and stimulated with insulin for 2 h. Liver nuclear extracts were immunoblotted with antibodies against Snail1 or lamin A/C. Snail1 levels were normalized to lamin A/C levels ($n = 3$). **f** HepG2 cells were pretreated with palmitate overnight and then stimulated with insulin. Cell extracts were immunoblotted with antibodies against Snail1 or Hsp90. Snail1 levels were quantified and normalized to Hsp90 levels ($n = 4$). Data are presented as mean ± SEM. *$p < 0.05$, two-tailed unpaired Student's $t$ test

$Snail1^{\Delta hep}$ mice (Fig. 3b). To extend these findings in obesity, we placed $Snail1^{\Delta hep}$ and $Snail1^{flox/flox}$ mice on a HFD for 10 weeks. Hepatocyte lipid droplets were substantially more abundant in $Snail1^{\Delta hep}$ mice relative to $Snail1^{flox/flox}$ littermates (Fig. 3c). Liver TAG levels were significantly higher in $Snail1^{\Delta hep}$ mice (Fig. 3d), although body weight and plasma TAG and NEFA levels were comparable between $Snail1^{\Delta hep}$ and $Snail1^{flox/flox}$ mice (Supplementary Fig. 3c, d).

To address concerns about the potential effect of tamoxifen on the observed phenotypes, we generated an independent line of $Snail1^{\Delta hep}$ mice by crossing $Snail1^{flox/flox}$ mice with albumin-Cre drivers. These mice were fed a HFD for 10 wks, and body weight was comparable between sex-matched $Snail1^{\Delta hep}$ and $Snail1^{flox/flox}$ mice (Supplementary Fig. 3e). Liver TAG levels were significantly higher in $Snail1^{\Delta hep}$ males and females relative to sex-matched $Snail1^{flox/flox}$ littermates (Supplementary Fig. 3f). Additionally, we deleted hepatocyte Snail1 in adult $Snail1^{flox/flox}$ mice using adeno-associated virus (AAV) Cre vectors[27]. $Snail1^{flox/flox}$ mice were fed a HFD for 6 weeks and then transduced with AAV-TBG-Cre or AAV-TBG-GFP (control) vectors. Snail1 was deleted in the livers but not the WAT of AAV-TBG-Cre-transduced mice (Supplementary Fig. 3g). Hepatocyte lipid droplets were more abundant and larger (Fig. 3e), and liver TAG levels were significantly higher in AAV-TBG-Cre mice relative to AAV-TBG-GFP mice (Fig. 3f). Body weight was similar between these two groups (Supplementary Fig. 3h). Furthermore, deletion of hepatocyte Snail1 similarly exacerbated fructose-induced liver steatosis in $Snail1^{\Delta hep}$ mice relative to $Snail1^{flox/flox}$ mice (Fig. 3g, h). Taken together, these data suggest that suppression of the noncanonical Snail1 arm exacerbates NAFLD independently of changes in body weight.

**Liver-specific expression of Snail1 protects against NAFLD.** We reasoned that enhancing the hepatic insulin/Snail1 arm might inhibit liver lipogenesis and NAFLD progression. C57BL/6 mice were fed a HFD for 7 weeks and transduced with Snail1 (GFP as control) adenoviral vectors. Recombinant Snail1 was detected in the liver but no other tissues as expected (Fig. 4a). Although body weight was comparable between Snail1 and GFP adenoviral-transduced mice (Fig. 4b), hepatocyte lipid droplets were considerably smaller and less abundant in Snail1 adenoviral-transduced mice (Fig. 4c). Liver TAG levels were significantly lower in Snail1 than in GFP mice (Fig. 4d). Likewise, liver-specific overexpression of Snail1 attenuated liver steatosis in ob/ob mice with genetic obesity (Fig. 4e, f). These data suggest that the insulin/Snail1 arm puts a brake on lipogenesis, thereby protecting against liver steatosis and lipotoxicity.

**Hepatic Snail1 suppresses liver lipogenesis in vivo.** To confirm inhibition of lipogenic programs by the Snail1 arm in vivo, we measured hepatic levels of key lipogenic enzymes Fasn, Acc1, and

Acl in $Snail1^{\Delta hep}$ mice fed a chow diet. Deletion of hepatic Snail1 substantially increased both mRNA (Fig. 5a) and protein (Fig. 5b) levels of these enzymes in $Snail1^{\Delta hep}$ mice compared to $Snail1^{flox/flox}$ littermates. To extend these findings to NAFLD, we placed $Snail1^{flox/flox}$ mice on a HFD for 6 weeks, followed by transduction with AAV-TBG-Cre (delete hepatic Snail1) or AAV-TBG-GFP (control) vectors. Deletion of hepatocyte Snail1 markedly increased both mRNA and protein levels of liver Fasn, Acc1, and Acl (Fig. 5c, d). We also performed an unbiased gene expression profiling analysis, and found that expression of lipogenic genes as well as genes encoding lipid droplet proteins was higher in mice with hepatocyte-specific ablation of Snail1 (Supplementary Fig. 4a). In contrast, expression of the genes that control lipid uptake, fatty acid β oxidation, and VLDL secretion was relatively normal (Supplementary Fig. 4a). Consistently, liver-specific overexpression of Snail1 did not alter expression of the genes controlling β oxidation and VLDL secretion (Supplementary Fig. 4b). Thus, protection against NAFLD by hepatic Snail1 can be explained, at least in part, by reduction in liver de novo lipogenesis. To further test this notion, C57BL/6 mice were fed a HFD for 6 weeks (increasing baseline lipogenesis) and transduced with Snail1 (overexpressing Snail1 in the liver) or GFP (control) adenoviral vectors. Overexpression of Snail1 profoundly decreased both mRNA and protein levels of Fasn, Acc1, and Acl in the liver (Fig. 5e, f). In contrast, expression of the genes that control liver inflammation ($TNF\alpha$, $IL6$, $MCP1$, $F4/80$), fibrosis ($Fn1$, vimentin, $\alpha SMA$, $Colla\ 1a1$), and EMT ($Ecad$, $Cldh1$) were similar between these two groups (Supplementary Fig. 4b).

Given Srebp-1c mediating insulin stimulation of lipogenesis, we tested if Snail1 counteracts Srebp-1c action. Deletion of hepatic Snail1 upregulated Srebp-1c in the liver; conversely, liver-specific overexpression of Snail1 downregulated hepatic Srebp-1c (Supplementary Fig. 5a, b). Srebp-1c potently stimulated Fasn promoter activity, as assessed by Fasn luciferase reporter assays; in contrast, Snail1 suppressed Fasn promoter activity (Supplementary Fig. 5c). Deletion of the serum response element (SRE), referred to as $Fasn(\Delta SRE)$, completely abolished Srebp-1c-stimulated activation of $Fasn(\Delta SRE)$; in contrast, Snail1 still suppressed $Fasn(\Delta SRE)$ activity (Supplementary Fig. 5d). Furthermore, Snail1 decreased Srebp-1c-stimulated activation of the Fasn promoter (Supplementary Fig. 5e). These data further support the concept that the noncanonical Snail1 arm counterbalances the canonical Srebp-1c lipogenic arm in response to insulin.

**Snail1 suppresses lipogenesis by an epigenetic mechanism.** We next sought to interrogate the genomic mechanism of Snail1 action, using the Fasn gene as a prototype of Snail1 targets. In mouse liver, Snail1 directly bound to the Fasn promoter as detected by chromatin immunoprecipitation (ChIP) (Fig. 6a). Snail1 directly inhibited Fasn promoter activity in HepG2 cells (Fig. 6b). These data further confirm that Snail1 is a transcriptional repressor of lipogenic genes.

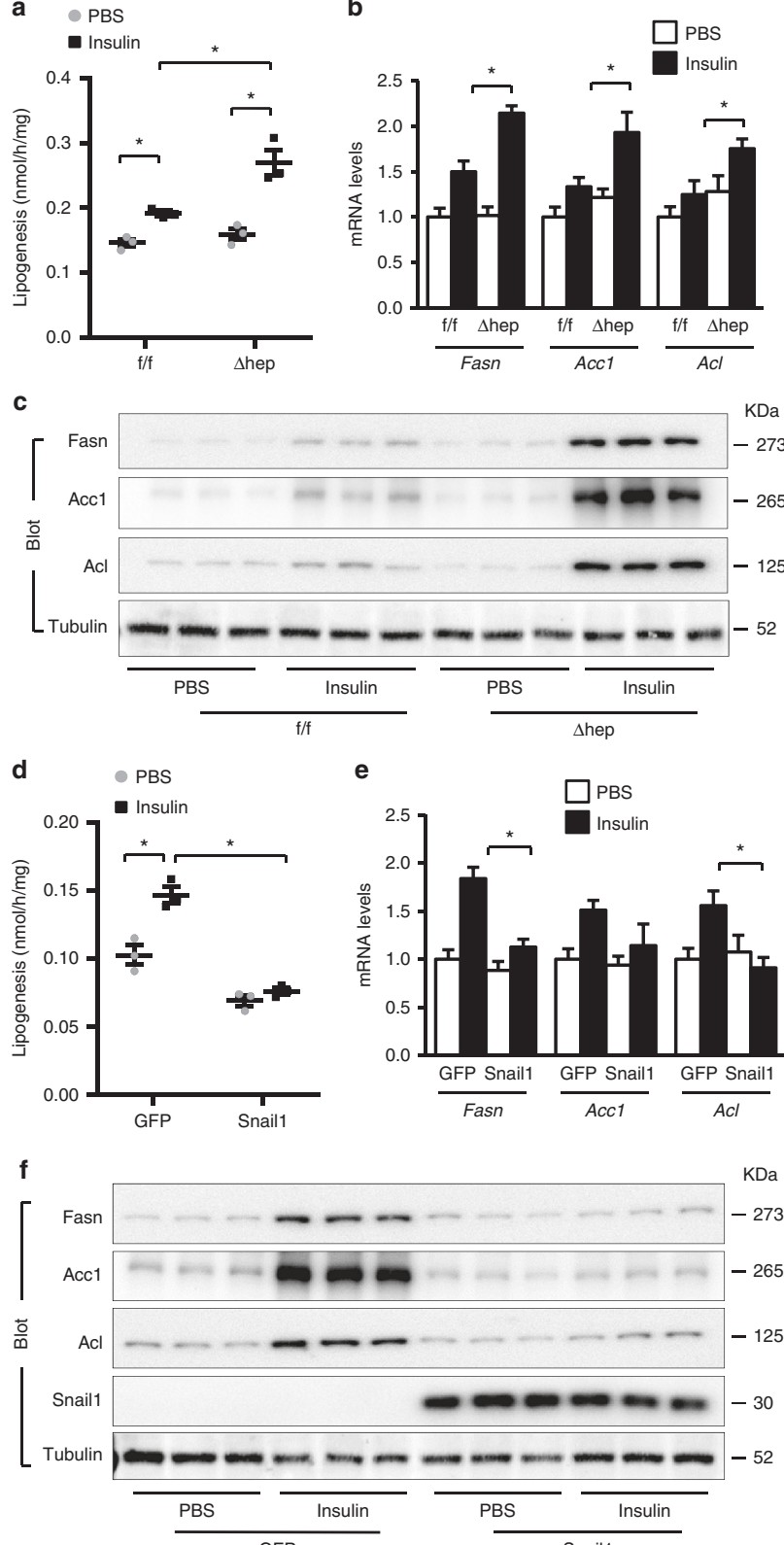

**Fig. 2** Insulin/Snail1 axis suppresses de novo lipogenesis in hepatocytes. **a–c** Primary hepatocytes were isolated from *Snail1^Δhep* and *Snail1^flox/flox* littermates and stimulated with insulin (50 nM) or PBS for 12 h (**a**, **c**) or 4 h (**b**). **a** De novo lipogenesis assays (normalized to protein levels; $n = 3$). **b** *Fasn*, *Acc1*, and *Acl* mRNA levels (normalized to 36B4 levels; $n = 4$). **c** Cell extracts were immunoblotted with the indicated antibodies. **d–f** Primary hepatocytes were transduced with Snail1 or GFP adenoviral vectors and stimulated with insulin (50 nM) for 12 h (**d**, **f**) or 4 h (**e**). **d** De novo lipogenesis (normalized to protein levels; $n = 3$). **e** *Fasn*, *Acc1*, and *Acl* mRNA levels (normalized to 36B4 levels; $n = 4$). **f** Cell extracts were immunoblotted with the indicated antibodies. Data are presented as mean ± SEM. *$p < 0.05$ by two-tailed unpaired Student's *t* test

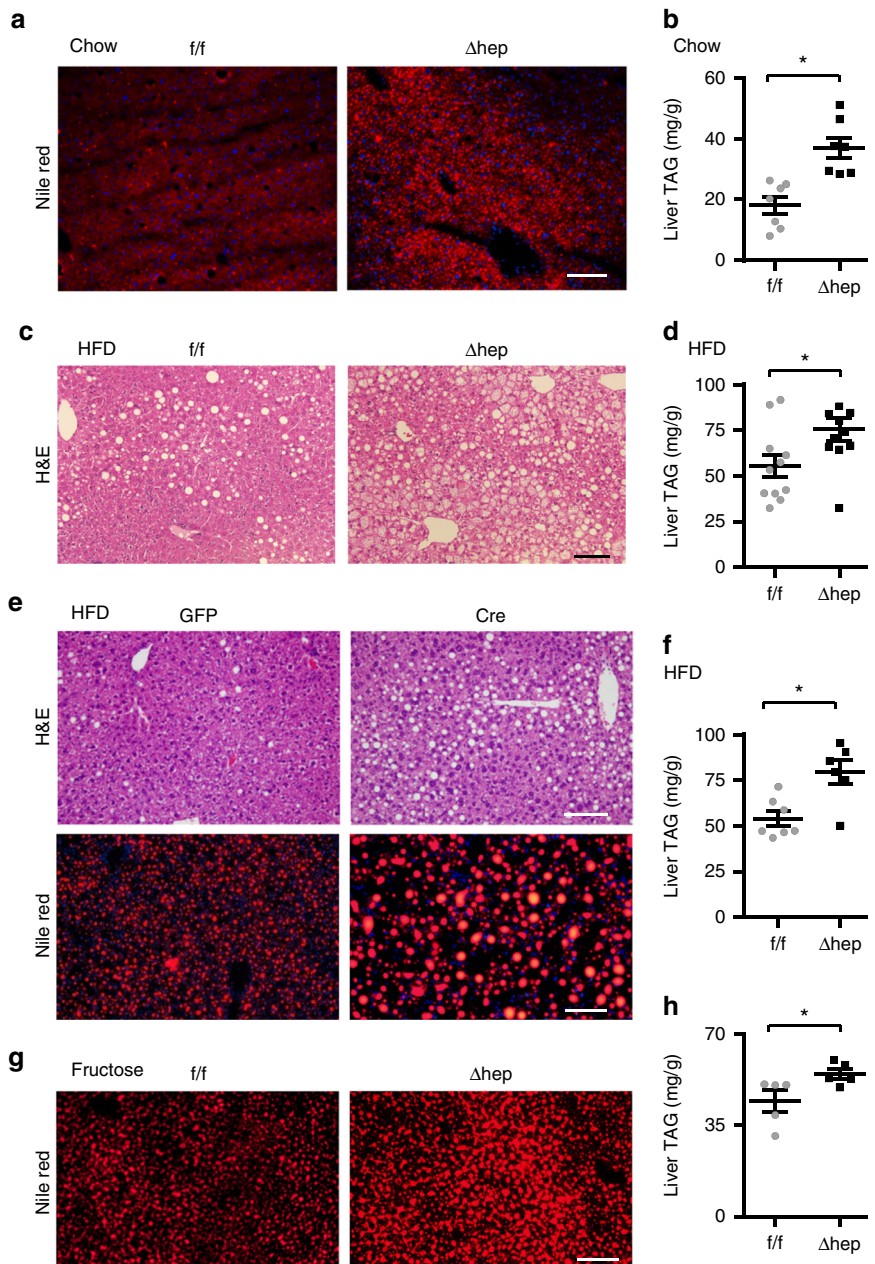

**Fig. 3** Hepatocyte-specific deletion of *Snail1* promotes NAFLD. **a**, **b** *Snail1^flox/flox* (*n* = 7) and *Snail1^Δhep* (*n* = 7) males (18 weeks) were fed a normal chow diet. Livers were isolated under non-fasted conditions. **a** Representative Nile red staining of liver sections. **b** Liver TAG levels (normalized to liver weight). **c**, **d** *Snail1^flox/flox* (*n* = 11) and *Snail1^Δhep* (*n* = 11) male littermates were fed a HFD for 10 weeks. **c** Representative H&E staining of liver sections. **d** Liver TAG levels (normalized to liver weight). **e**, **f** *Snail1^flox/flox* males were fed a HFD for 6 weeks and transduced with AAV-TBG-GFP (*n* = 7) or AAV-TBG-Cre (*n* = 6) vectors for 4 weeks. **e** Representative H&E or Nile red staining of liver sections. **f** Liver TAG levels (normalized to liver weight). **g**, **h** *Snail1^flox/flox* (*n* = 5) and *Snail1^Δhep* (*n* = 5) males were fed a fructose diet for 10 weeks. Scale bars: 100 μm. Data are represented as mean ± SEM. *$p < 0.05$, two-tailed unpaired Student's *t* test

HDAC1 and HDAC2 have been known to repress expression of target genes by deacetylating histone H3 lysine-9 (H3K9) and H3K27, critical epigenetic modifications. We found that Snail1 coimmunoprecipitated with both HDAC1 and HDAC2 in mouse liver (Fig. 6c). Hence, we measured acetylation of H3K9 (H3K9ac) and H3K27 (H3K27ac) on the *Fasn* promoter. Liver-specific overexpression of Snail1 significantly reduced both H3K9ac and H3K27ac levels in the liver (Fig. 6d). In contast, Snail1 did not alter the levels of H3K9ac and H3K27ac on unrelated *NCH* and *Actb* promoters (Fig. 6d). To extend these findings to endogenous Snail1, we assessed H3K9ac and H3K27ac

levels in the livers of *Snail1^Δhep* mice. Ablation of hepatic *Snail1* markedly increased both H3K9ac and H3K27ac levels on the *Fasn* promoter in the livers of *Snail1^Δhep* mice relative to *Snail1^flox/flox* littermates (Fig. 6e). Notably, other forms of histone modifications on the *Fasn* promoter, including trimethylation of H3K4, H3K9, and H3K27, were not altered by either overexpression or ablation of Snail1 in the liver (Supplementary Fig. 6a, b).

To confirm inhibition of lipolysis by Snail1-elicited histone deacetylation, primary hepatocytes were transduced with Snail1 and treated with trichostatin A (TSA), a selective HDAC inhibitor. TSA blocked the ability of Snail1 to inhibit lipogenesis

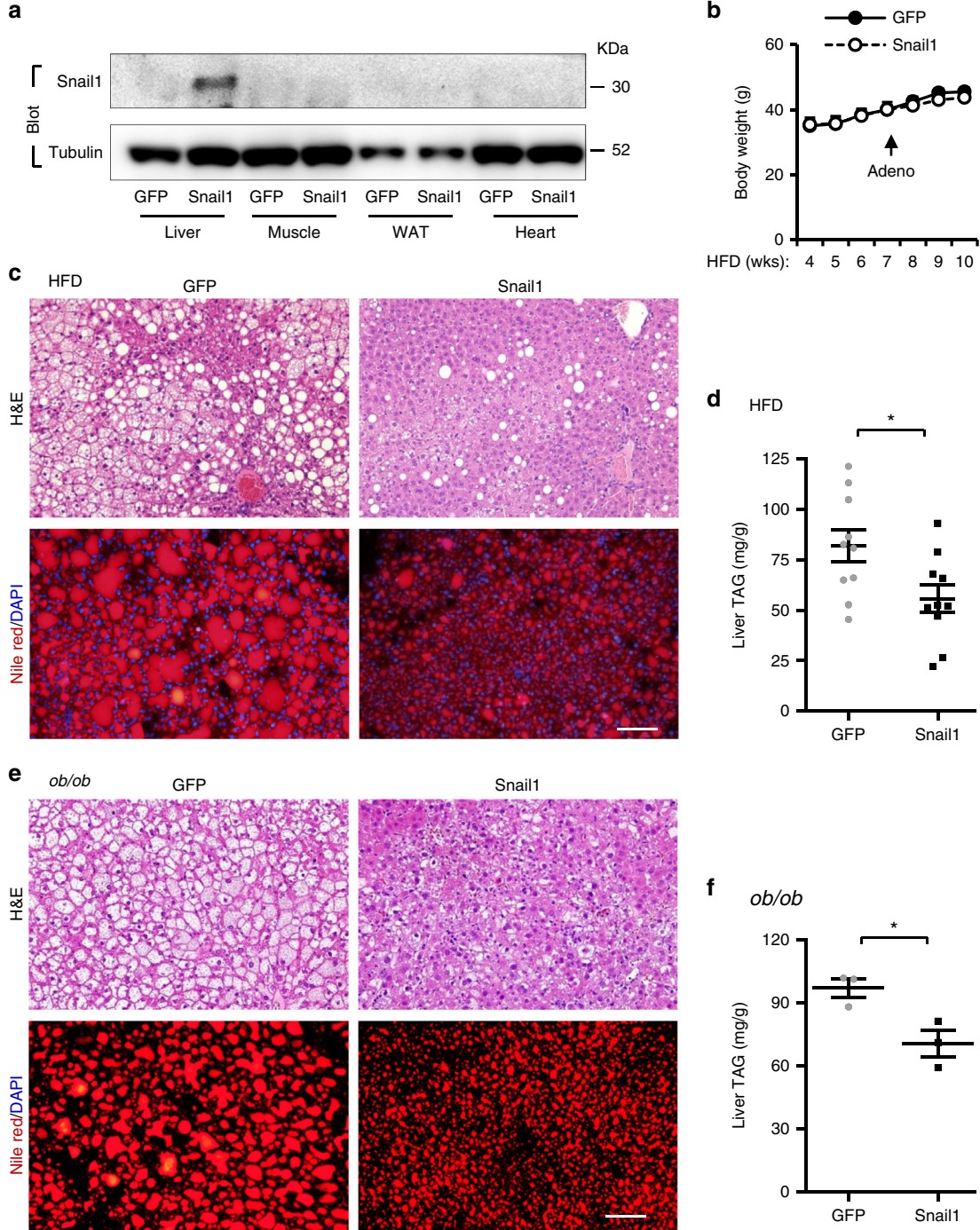

**Fig. 4** Liver-specific overexpression of Snail1 protects against NAFLD. **a–d** C57BL/6 male mice (8–9 weeks) were fed a HFD for 7 weeks and transduced with Snail1 ($n = 10$) or GFP ($n = 10$) adenoviral vectors. Livers were harvested 3 weeks after transduction. **a** Tissue extracts were immunoblotted with the indicated antibodies. **b** Growth curves. **c** Representative H&E or Nile red staining of liver sections. Scale bar: 100 μm. **d** Liver TAG levels (normalized to liver weight). **e, f** *ob/ob* male mice (9 weeks) were transduced with GFP ($n = 3$) or Snail1 ($n = 3$) adenoviral vectors for 3 weeks. **e** Representative H&E or Nile red staining of liver sections. **f** Liver TAG levels (normalized to liver weight). Scale bars: 100 μm. Data are presented as mean ± SEM. *$p < 0.05$, two-tailed unpaired Student's *t* test

(Supplementary Fig. 6c). TSA also reversed Snail1-induced suppression of *Fasn* expression (Supplementary Fig. 6d, e). To corroborate these findings, we deleted the SNAG domain of Snail1 (amino acids 1–20) responsible for binding to HDAC1/2, referred to as ΔN20. Primary hepatocytes were transduced with Snail1 or ΔN20 adenoviral vectors, followed by stimulation with insulin. ΔN20, unlike Snail1, was unable to suppress insulin

stimulation of *Fasn* expression (Supplementary Fig. 6f) and lipogenesis (Fig. 6f). Importantly, ΔN20 also lost the ability to elicit deacetylation of H3K9 (Supplementary Fig. 6g) and H3K27 (Fig. 6g). Collectively, these data suggest that the insulin/Snail1 axis suppresses de novo lipogenesis through epigenetic reprogramming of lipogenic genes, particularly via deacetylating H3K9 and H3K27.

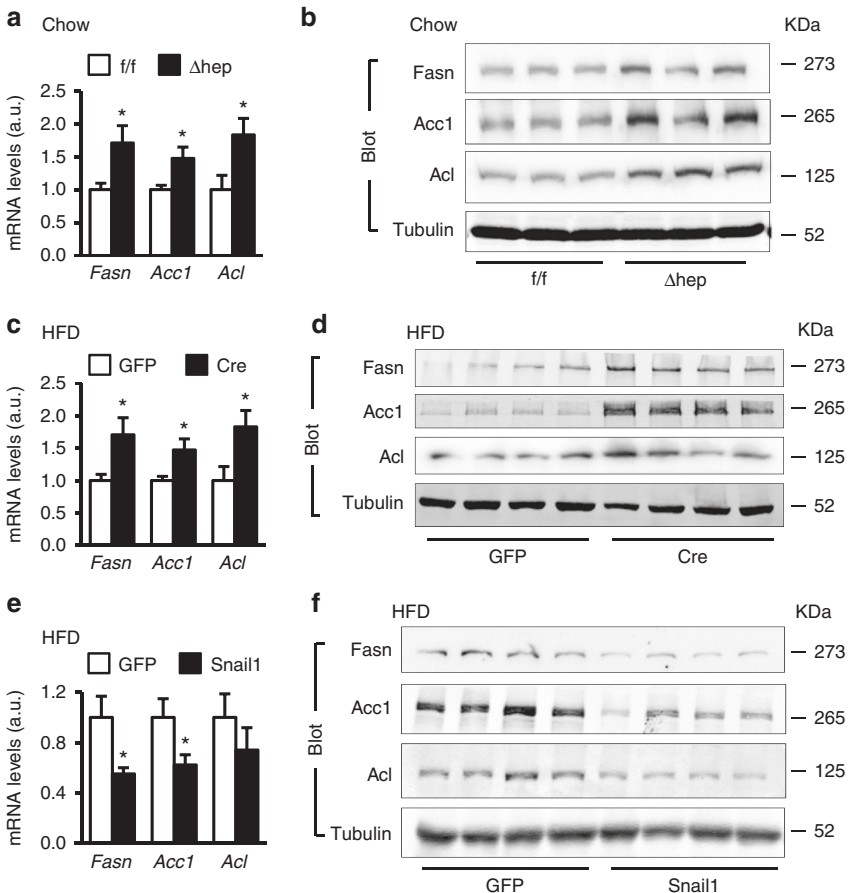

**Fig. 5** Snail1 directly suppresses liver lipogenic programs in vivo. **a**, **b** Snail1$^{\Delta hep}$ ($n = 6$) and Snail1$^{flox/flox}$ ($n = 6$) males fed a normal chow diet for 18 weeks. **c**, **d** Snail1$^{flox/flox}$ male mice (8–9 weeks) were fed a HFD for 6 weeks and transduced with AAV-TBG-GFP ($n = 6$) or AAV-TBG-Cre ($n = 6$) vectors. Livers were harvested 4 weeks after AAV transduction. **e**, **f** C57BL/6 male mice (8–9 weeks) were fed a HFD for 7 weeks and transduced with Snail1 ($n = 5$) or GFP ($n = 5$) adenoviral vectors. **e** Liver Fasn, Acc1, and Acl expression was assessed by qPCR (normalized to 36B4 expression). **f** Liver extracts were immunoblotted with the indicated antibodies. Fasn, Acc1, and Acl levels were quantified and normalized to α-tubulin levels. Data are presented as mean ± SEM. *$p < 0.05$, two-tailed unpaired Student's t test

**Ablation of hepatic Snail1 exacerbates insulin resistance.** Liver steatosis has been thought to induce insulin resistance, prompting us to examine glucose metabolism in Snail1$^{\Delta hep}$ mice. Plasma insulin levels was normal in Snail1$^{\Delta hep}$ mice fed a normal chow diet (Snail1$^{\Delta hep}$: 0.70 ± 0.07 ng/ml, $n = 5$; Snail1$^{flox/flox}$: 0.61 ± 0.09 ng/ml, $n = 5$; p = 0.42). We placed Snail1$^{\Delta hep}$ and Snail1$^{flox/flox}$ littermates on a HFD for 10 weeks. Body weight was similar between these two groups (Supplementary Fig. 3c). Snail1$^{\Delta hep}$ mice developed more severe hyperinsulinemia, a hallmark of systemic insulin resistance, relative to Snail1$^{flox/flox}$ littermates (Fig. 7a). To confirm insulin resistance, we performed insulin (ITT) and glucose (GTT) tolerance tests. Insulin had reduced ability to decrease blood glucose in Snail1$^{\Delta hep}$ mice compared to Snail1$^{flox/flox}$ mice (Fig. 7b). The areas under the curves (AUC) of both ITT and GTT were significantly higher in Snail1$^{\Delta hep}$ mice (Fig. 7b, c). Furthermore, the levels of insulin-stimulated phosphorylation of Akt (pSer473 and pThr308) in the liver was significantly lower in Snail1$^{\Delta hep}$ than in Snail1$^{flox/flox}$ littermates (Fig. 7d). Thus, Snail1$^{\Delta hep}$ mice are prone to both liver steatosis and insulin resistance.

To determine whether liver-specific overexpression of Snail1 has the opposite effect, C57BL/6 mice were fed a HFD for 7 weeks and transduced with Snail1 or GFP (control) adenoviral vectors. Overexpression of Snail1 significantly attenuated HFD-induced insulin resistance and glucose intolerance (Fig. 7e, f), although

body weight was similar between Snail1 and GFP adenoviral-transduced mice (Fig. 4b). Concurrently, hepatic Snail1 substantially decreased liver steatosis in obese mice (Fig. 4c). Together, these data suggest that the hepatic insulin/Snail1 axis ameliorates insulin resistance in obesity, presumably through decreasing liver steatosis.

## Discussion

In this study, we have identified hepatic Snail1 as an unrecognized suppressor of lipogenesis. Snail1 bound to the promoters of lipogenic genes where it recruited HDAC1/2 to catalyze repressive deacetylation of H3K9 and H3K27. Insulin robustly upregulated hepatic Snail1 in both primary hepatocytes and livers, defining an unrecognized noncanonical insulin/Snail1 pathway that epigenetically suppresses lipogenesis.

In hepatocytes, insulin substantially increased both expression and stability of Snail1 in a PI 3-kinase-dependent manner, defining an insulin-PI 3-kinase-Snail1 cascade. Snail1 overexpression blocked insulin-stimulated expression of lipogenic enzymes and lipogenesis in primary hepatocytes. Liver-specific overexpression of Snail1 suppressed the hepatic lipogenic program and protected against dietary NAFLD. Conversely, ablation of hepatic Snail1, using three distinct approaches (tamoxifen, AAV-TBG-Cre, and albumin-Cre), had the opposite effect. These

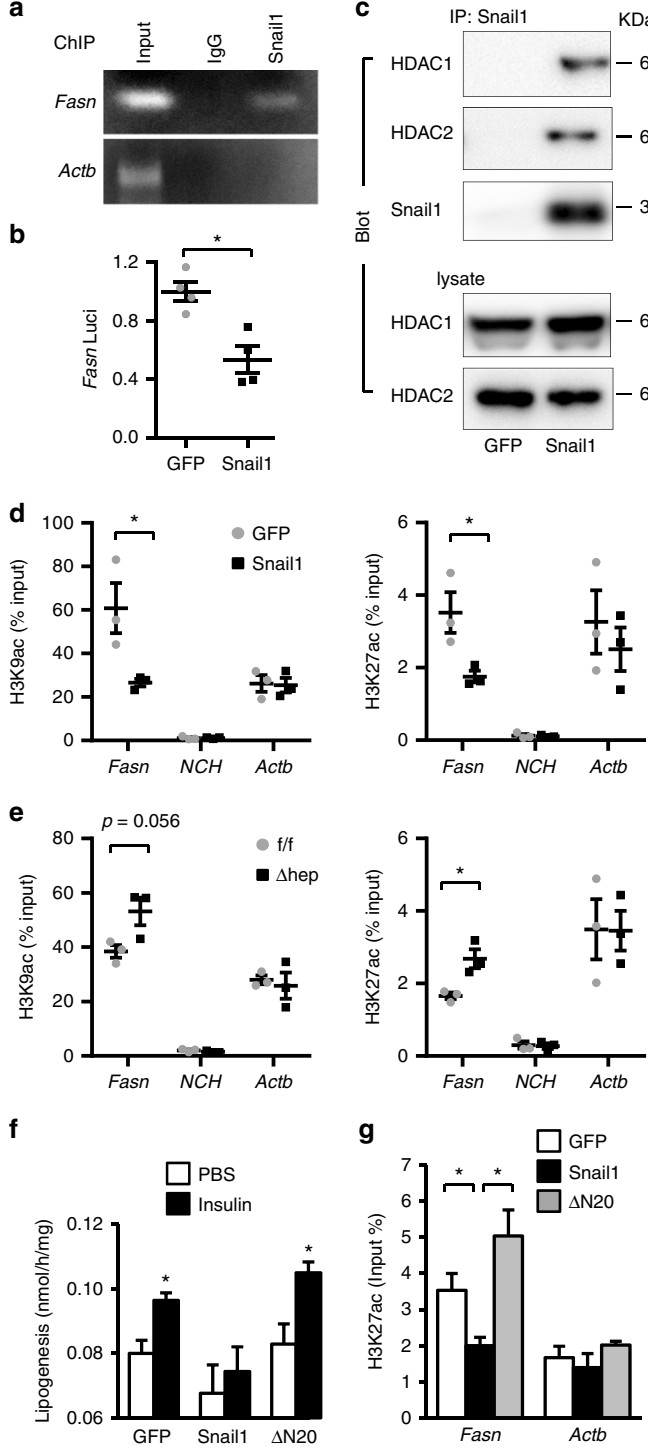

**Fig. 6** Snail1 epigenetically represses the *Fasn* promoter. **a** Male mice were transduced with Snail1 adenoviral vectors for 3 weeks. Binding of Snail1 to the *Fasn* promoter was assessed in the liver by ChIP. **b** *Fasn* luciferase activity (normalized to β-gal internal controls). **c** Male mice were transduced with Snail1 adenoviral vectors for 3 weeks. Liver extracts were immunoprecipitated with antibody against Snail1 and immunoblotted with antibodies against HDAC1 or HDAC2. **d** C57BL/6 male mice (8–9 weeks) were fed a HFD for 7 weeks and then transduced with Snail1 ($n = 3$) or GFP ($n = 3$) adenoviral vectors. Livers were harvested 3 weeks after transduction. The levels of liver H3K9ac and H3K27ac on the *Fasn*, *NCH*, or *Actb* promoter were measured by ChIP-qPCR and normalized to inputs. **e** Snail1$^{\Delta hep}$ ($n = 3$) and Snail1$^{flox/flox}$ ($n = 3$) males were fed a HFD for 10 weeks. The levels of liver H3K9ac or H3K27ac on the *Fasn*, *NCH*, or *Actb* promoter were measured by ChIP. **f**, **g** Primary hepatocytes were transduced with Snail1, ΔN20, or GFP adenoviral vectors and stimulated with or without insulin (50 nM) for 12 h. **f** Lipogenesis. **g** H3K27ac levels on the *Fasn* promoter. Data are presented as mean ± SEM. *$p < 0.05$, two-tailed unpaired Student's *t* test

mentioning that obesity is associated with compensatory hyper-insulinemia in part due to adipose insulin resistance. Hyper-insulinemia further drives hepatic lipogenesis, particular when liver is still sensitive to insulin and/or the Snail1 arm is dysfunctional, thereby exacerbating NAFLD.

Mechanistically, we found that Snail1 directly bound to the *Fasn* promoter and repressed *Fasn* promoter activity. Snail1 has been known to bind, via its SNAG domain, to many epigenetic enzymes, including HDAC1/2, LSD1, Ezh2, and G9a[15,16,32]. We confirmed that Snail1 bound to HDAC1/2 in the liver. Overexpression of Snail1 decreased, whereas ablation of Snail1 increased, the levels of H3K9ac and H3K27ac, active epigenetic marks, on the *Fasn* promoter. Remarkably, inhibition of HDAC1/2 by TSA blocked Snail1-induced suppression of lipogenesis. Deleting the SNAG domain also abrogated the ability of Snail1 to inhibit insulin stimulation of lipogenesis. These findings support the notion that Snail1 suppresses lipogenesis at least in part by recruiting HDAC1/2 to lipogenic gene promoters where HDAC1/2 catalyze repressive deacetylation of H3K9 and H3/K27.

Given that liver steatosis is linked to insulin resistance, it is not unexpected that ablation of hepatic Snail1 exacerbated, whereas liver-specific overexpression of Snail1 ameliorated, HFD-induced insulin resistance and glucose intolerance. It is worth mentioning that simple liver steatosis has been reported to be dissociated from insulin resistance[33]. Diacylglycerol and several lipid species, rather than inert TAG, are able to induce insulin resistance[34–37]. In light of these observations, we propose that hepatic Snail1 may selectively inhibit production of toxic lipid species that induce insulin resistance. Additional studies are needed to test this hypothesis in the future.

Liver steatosis is considered, historically, as the first hit during NAFLD development[38]. In the presence of the secondary and/or multiple other hits (e.g., oxidative stress), relatively benign fatty liver progresses to pathogenic NAFLD/NASH manifested by liver inflammation, injury, and fibrosis[38,39]. The severity of liver steatosis is influenced by multiple metabolic pathways, which are controlled largely at the transcriptional level. Transcription factors Srebp-1c, ChREBP, and Lxr have been extensively characterized for their ability to stimulate de novo lipogenesis and liver steatosis[6,40–43]. Hepatic PPARγ, which is upregulated in obesity, stimulates expression of lipid droplet proteins as well as fatty acid transporters, thereby promoting liver steatosis[8,44–48]. In contrast, hepatic PPARα, PPARβ/δ, Foxa2, and circadian clock BMAL1 exert anti-steatosis action by promoting fatty acid β oxidation[49–54]. Recent studies have highlighted the impact of

findings demonstrate that insulin concomitantly stimulates both the canonical lipogenic (e.g., Srebp-1c) and the noncanonical Snail1 (anti-lipogenesis) pathways. The noncanonical Snail1 arm likely puts a brake on the canonical lipogenic arm, combating excessive lipogenesis and lipotoxicity. In obesity, the insulin/Snail1 brake was impaired, likely contributing to increased hepatic lipogenesis and NAFLD. Aside from insulin, Wnts, TGFβ1, and additional factors also upregulate Snail1[18,28,29]. Both Wnt and TGFβ1 suppress lipogenesis[30,31]. Therefore, Snail1 may serve as a common node upon which these factors converge to regulate lipogenesis and intracellular lipid content. It is worth

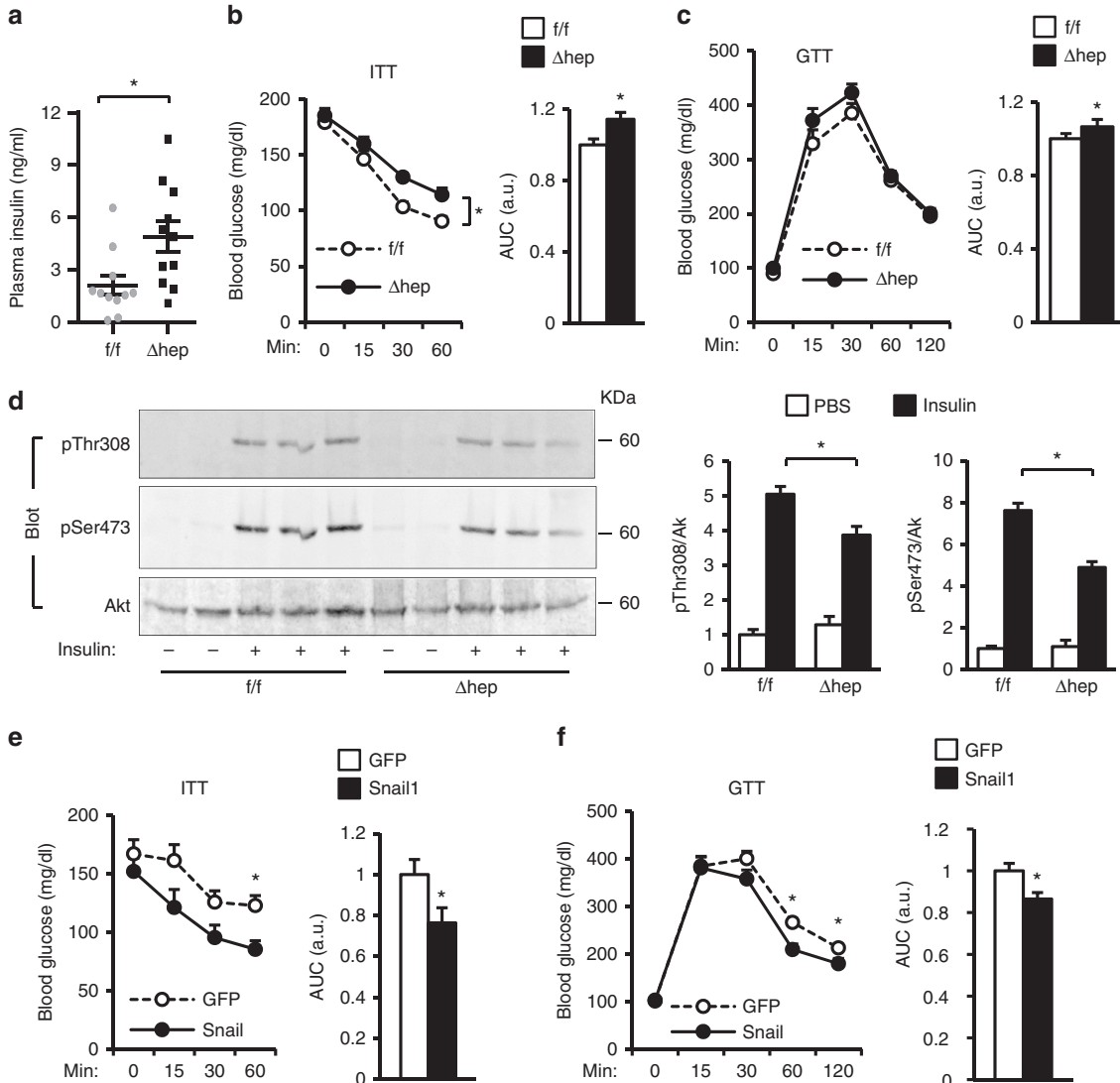

**Fig. 7** Hepatic Snail1 ameliorates insulin resistance in obesity. **a–d** Snail1$^{\Delta hep}$ ($n = 11$) and Snail1$^{flox/flox}$ ($n = 11$) male mice (8–9 weeks) were fed a HFD for 10 weeks. **a** Overnight fasting plasma insulin levels. **b** ITT. AUC areas under curves. **c** GTT. **d** Mice were fasted overnight (14 h) and injected with insulin (0.5 units/kg). Liver extracts were prepared 5 min later and immunoblotted with anti-phospho-Akt (pSer473 or pThr308) or anti-Akt antibodies. Akt phosphorylation was normalized to total Akt levels. **e**, **f** C57BL/6 male mice (8–9 weeks) were fed a HFD for 7 weeks and transduced with Snail1 ($n = 10$) or GFP ($n = 10$) adenoviral vectors for 2 weeks after transduction. **e** ITT. **f** GTT. Data are presented as mean ± SEM. *$p < 0.05$, two-tailed unpaired Student's $t$ test

negative regulators of lipogenesis on NAFLD. Fxr, nuclear receptors for bile acids, suppresses lipogenesis and liver steatosis[55–57]. Circadian clock Rev-erbα/β also suppresses lipogenesis, and liver-specific ablation of rev-erbα/β results in NAFLD[58,59]. Notably, in contrast to genetic lipogenic programs that have been extensively investigated, epigenetic regulation of lipid metabolism remains largely unclear. In this regard, Snail1-elicited epigenetic reprogramming of hepatic lipogenesis, identified in this study, likely points to a new direction in the NAFLD field.

In conclusion, we have unveiled the noncanonical insulin-Snail1 pathway that epigenetically suppresses hepatic lipogenesis. Impairment in this pathway increases hepatic de novo lipogenesis, contributing to NAFLD in obesity. Our data suggest that the insulin/Snail1/epigenetic axis may serve as a potential therapeutic target for the treatment of NAFLD and metabolic disease.

## Methods

**Animal treatments**. Animal experiments were conducted following the protocols approved by the University of Michigan Institutional Animal Care and Use

Committee (IACUC). Snail1$^{flox/flox}$ and albumin-CreER$^{T2}$ mice (C57BL/6 background) have been characterized previously[25,26]. Mice were housed on a 12-h light-dark cycle in the Unit for Laboratory Animal Medicine at the University of Michigan (ULAM), and fed ad libitum either a normal chow diet (9% fat in calories; TestDiet, St. Louis, MO), a HFD (60% fat in calories; Research Diets, New Brunswick, NJ), or a fructose diet (60% Fructose; in calories; Research Diets, New Brunswick, NJ).

To generate mice hepatocyte-specific ablation of Snail1, Snail1$^{flox/flox}$ mice were crossed with albumin-CreER$^{T2}$ or albumin-Cre drivers. Snail1$^{flox/flox}$;albumin-CreER$^{T2}$ mice were intraperitoneally injected with tamoxifen (Cayman) (0.5 mg/mouse, twice 2 days apart) to obtain Snail1$^{\Delta hep}$ mice. Snail1$^{flox/flox}$ mice were similarly treated with tamoxifen as control. Alternatively, adult Snail1$^{flox/flox}$ mice were fed a HFD for 6–7 wks and injected with AAV-TBG-Cre or AAV-TBG-GFP vectors via tail veins (10$^{11}$ viral particles/mouse). To generate mice with liver-specific overexpression of Snail1, C57BL/6 male mice (12 wks) were fed a HFD for 6 weeks and then transduced with Snail1 or GFP (control) adenoviral vectors via tail vein injection (10$^{11}$ viral particles/mouse). ob/ob mice (9 wks) were similarly transduced with Snail1 or GFP adenoviral vectors.

To measure liver Snail1 levels, C57BL/6 males (8–10 wks) were fed a HFD or chow diet for 6 weeks, fasted overnight (14 h), and stimulated with insulin (0.75 units/kg body weight, i.p.) for 2 h. Liver nuclear extracts were prepared and immunoblotted with antibodies against Snail1 or lamin A/C (loading control).

**Glucose (GTT) and insulin (ITT) tolerance tests**. Mice were fasted overnight (GTT) or for 6 h (ITT), and intraperitoneally injected with glucose (1 g/kg body weight) for GTT or human insulin (1 unit/kg body weight) for ITT. Blood glucose was measured in tail veins 0, 15, 30, 60, and 120 min after injection. Plasma insulin levels were measured using mouse insulin ELISA kits (CRYSTAL CHEM, Downers Grove, IL).

**Nile red staining and liver TAG levels**. Liver frozen sections were fixed with 4% paraformaldehyde for 20 min, washed twice with PBS, stained with Nile red (1 μg/ml in PBS) for ~30 min, washed twice with PBS, and visualized using fluorescent microscope. Liver samples were homogenized in 1% acetic acid, and lipids were extracted using 80% chloroform/Methanol (2:1). Organic fractions were dried in a chemical hood, resuspended in a KOH (3 M)/ethanol solution, incubated at 70 °C for 1 h, mixed with $MgCl_2$ (0.75 M), and centrifuged. Aqueous fractions were used to measure TAG levels using Free Glycerol Reagent (Sigma).

**Immunoblotting and immunoprecipitation**. Tissues or cells were homogenized in a lysis buffer (50 mM Tris HCl, pH 7.5, 1.0% NP-40, 150 mM NaCl, 2 mM EGTA, 1 mM $Na_3VO_4$, 100 mM NaF, 10 mM $Na_4P_2O_7$, 1 mM PMSF, 10 mg/ml aprotinin, and 10 mg/ml leupeptin). Tissue or cell extracts were immunoprecipitated and/or immunoblotted with the indicated antibodies (Supplementary table 1). Uncropped scans of important blots were provided in the Supplementary Figs. 7–9.

**Cell cultures and treatments**. HepG2 cells (ATCC) were grown in DMEM containing 5 mM glucose, 10% calf serum, 100 units/ml penicillin, and 100 μg/ml streptomycin at 5% $CO_2$ and 37 °C. Primary hepatocytes were isolated using liver perfusion with type II collagenase (Worthington Biochem, Lakewood, NJ), and were grown in William E Medium (Sigma, St. Louis, MO) supplemented with 2% FBS, 100 units/ml penicillin, and 100 μg/ml streptomycin.

HepG2 cells were deprived of serum overnight, pretreated with wortmannin (100 nM) or MK2066 (100 nM) for 30 min, and then stimulated with insulin (50 nM) for 2 h. Additionally, HepG2 cells were pretreated with palmitate (100 μM) overnight and then stimulated with insulin (100 nM) for 2 h. Cell extracts were prepared for various assays.

In separate cohorts, HepG2 cells were transfected with Snail1 plasmids. Forty-eight hour later, the cells were derived of serum overnight, and were then treated with cycloheximide (5 μg/ml) in the presence of either insulin (100 nM) or PBS (control) for 0–8 h. Cell extracts were immunoblotted with antibodies against Snail1 or α-tubulin. To calculate half-life, Snail1 protein was quantified and normalized to α-tubulin levels. Snail1 abundance was presented as a ratio to its baseline levels prior to cycloheximide treatment, and was plotted against durations of cycloheximide treatment. In separate experiments, HepG2 cells were transduced with Snail1 adenoviral vectors for 24 h, and then stimulated with insulin (100 nM) with or without MG132 (5 μM) for 2 h. Cell extracts were immunoprecipitated with antibody against Snail1 and immunoblotted with antibodies against ubiquitin or Snail1.

Primary hepatocytes were transduced with Snail1 or GFP adenoviral vectors (1000 viral particles per cell) for 24 h. Hepatocytes were deprived of serum overnight in DMEM supplemented with 5 mM glucose, and then grown in DMEM with 22.5 mM glucose in the presence or absence of insulin (50 nM) for additional 4 h or 12 h. Cell extracts were prepared for various assays.

**Fasn luciferase reporter assays**. The rattus Fasn promoter (from −225 to +45) was prepared by PCR (forward primer: 5′-AGTGCCTCTCATGTATGCTTAA-3′, and reverse primer: 5′-TCCCGCAGTCTCGATACCTTGG-3′) and inserted into pGL3 vectors. HepG2 cells were co-transfected with Fasn, or Fasn (ΔSRE), luciferase reporter plasmids and the indicated vectors[60]. Luciferase activities were measured 72 h after transfection using a kit (Progema, Madison, WI), and were normalized to β-gal levels (internal control).

**De novo lipogenesis**. Primary hepatocytes were transduced with the indicated adenoviral vectors as described previously[61], and concomitantly treated with or without TSA (2 μM). De novo lipogenesis was assessed and normalized to total protein levels as described previously[62]. To assess insulin stimulation of lipogenesis, hepatocytes were deprived of serum overnight in the presence of 5 mM glucose, and stimulated with 50 nM insulin for 12 h prior to lipogenesis assays.

**Quantitative real time RT-PCR (qPCR)**. Total RNAs were extracted using TRIzol reagent (Invitrogen life technologies, Carlsbad, CA). The first-strand cDNAs were synthesized using random primers and M-MLV reverse transcriptase (Promega, Madison, WI). qPCR was performed using Absolute QPCR SYBR Mix (Thermo Fisher Scientific, UK) and Mx3000P real time PCR system (Stratagene, LA Jolla, CA). qPCR primers were listed in supplementary table 2.

**Chromatin immunoprecipitation (ChIP)**. ChIP assays were described previously[11]. Briefly, liver samples were treated with 1% formaldehyde for 10 min to crosslink DNA-protein complexes. Genomic DNA was extracted, and sheared to 200–500 bp fragments using a sonicator (Model Q800R, QSONICA). DNA-protein complexes were immunoprecipitated with the indicated antibodies (Supplementary table 1). Crosslink was reversed by heating at 65 °C for 4 h. DNA was purified and used for PCR or qPCR analysis. Primers flanking the putative Snail1-binding motifs of the Fasn promoter were listed in supplementary table 2.

**Statistical analysis**. Differences between two groups were analyzed by two-tailed Student's t test. Longitudinal data (growth curves, GTT and ITT) were further analyzed by ANOVA and Bonferroni posttest using Prism 7. $P < 0.05$ was considered statistically significant.

**Data availability**. All relevant data are available upon request.

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

## Acknowledgements

We thank Drs. Zheng Chen, Mark J. Canet, Hong Shen, Gong Cheng, Deqiang Zhang, Yin Lei, Xin Tong, and Bishr M. Omary for assistance and discussions. We thank Dr. Stephen J. Weiss (University of Michigan) for providing *Snail1*^*flox/flox*^ mice. This study was supported by grants DK094014, DK114220, and DK115646 (to L.R.) and DK095201 (to Y.S.) from the National Institutes of Health, American Heart Association Post-doctoral Fellowship 14POST20230007 (to Yan Liu), and National Natural Science Foundation of China Grant 81420108006 (to Yong Liu). This work utilized the cores supported by the Michigan Diabetes Research and Training Center (NIH DK20572), the University of Michigan Nathan Shock Center (NIH P30AG013283), and the University of Michigan Gut Peptide Research Center (NIH DK34933).

## Author contributions

Yan Liu, L.J., C.S., and N.I. conducted the experiments, Yan Liu and L.R. designed the experiments and wrote the paper, and Yan Liu, L.J., C.S., N.I., Y.M.S., Yong Liu, and L.R. performed data analysis and edited the paper.

## Additional information

**Competing interests:** The authors declare no competing interests.

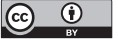

