## [Peer Review File · Nature Communications]

Reviewers' comments:

Reviewer #1 (Remarks to the Author):

This manuscript examines the effects of the zinc-finger binding protein Snail1 as a potential factor regulating lipogenesis in the liver. Snail1 had been implicated in the Endothelial-mesenchymal transition through its actions on e-cadherin.

The data presented here can be best interpreted as showing that hepatocyte lines and hepatocytes lacking Snail1 have significantly more insulin stimulated lipogenesis than cells that express Snail1.

Overall some of the data are interesting however they don't make for a coherent story.

It is also disappointing that the intro does not provide any information about Snail. To quote from a review written by Kaufhold, "Snail1-induced EMT involves the loss of E-cadherin and claudins with concomitant upregulation of vimentin and fibronectin, among other biomarkers." It seems important to also measure these markers in the models proposed here. If limitation of lipogenesis by Snail1 is offset by increased fibrosis this is an important point. Also it is known that Snail is regulated by PI3Kinase in other models, which does not seem to be noted here.

With regard to describing fatty liver, the Nile red staining is basically uninformative. It is not clear how many liver sections were examined to describe fatty liver and who examined them. Ideally this should be done by a hepatopathologist who is blinded to the nature of the samples. Thus inflammation and fibrosis can also be scored. Fatty liver can be quite heterogeneous so representative slices can be deceiving.

While inclusion of data from both ex-vivo and in-vivo models is helpful the switching back and forth within single figures makes the data appear less rather than more coherent. Also it's never entirely clear why transformed hepatocytes are used.

Specific Points:

1. The authors conflate NAFLD with NAFL. Fatty liver as in NAFL, which may or may not be benign may or may not progress to disease which includes NASH, fibrosis, cirrhosis and potentially development of HCC. In this report only fatty liver has been examined.
2. The authors refer to one of potentially 2 explanation for fatty liver in the context of insulin resistance which is that paradoxically hepatic insulin resistance is associated with increase lipogenesis which also implies that it is fatty liver that causes insulin resistance. Several recent reviews point out that there is likely differential sensitivity of liver in comparison to fat with regard to insulin action and that as fat becomes resistant serum insulin rises while the liver is still sensitive which is why fatty liver occurs. This should be included in the narrative
3. Given the potential importance of the association between Snail1 and insulin action, it would be important to have more than a single data point on this interaction. All that is shown is that after six weeks of a high fat diet, when mice are fasted overnight, Snail1 expression is induced by insulin at two hours only in mice fed chow. What happens in the fed state? Were dose response curves performed? If more insulin is given to HFD group will expression be induced?
4. The HFD is not an HFD – as per the methods the 60% fat diet administered also contains increased amounts of sucrose compared to chow. This should be noted.
5. Can doses of insulin and glucose be included in the methods section referring to GTT and ITT?
6. Based on the data shown Snail1 does not suppress lipogenesis, it limits insulin induced lipogenesis. This should be clarified. Does Snail1 deletion shift the dose response?
7. To reach the conclusion in Figure 4, that overexpression of Snail1 suppresses lipogenesis, livers of

mice with AAV mediated increased Snail1 expression are shown in animals fed a high fed diet. The data provided are a single H & E slice and overall liver triglycerides. This is not enough to show suppression of lipogenesis which ideally should be looking at fatty acid synthesis and expression of lipogenic enzymes in the liver. Other explanation for decreased liver fat are increased fatty acid oxidation or increased export.

8. The conclusion appears to be inferred from Figure 5 where deletion of hepatic Snail leads to increased expression of lipogenic enzymes and which is described as Snail directly suppressing liver lipogenesis. However what is shown in figure 5 is enhanced expression of lipogenic enzymes in the absence of snail which is different from showing suppression.

9. To be consistent similar data is needed from the model described in Figures 4 and 5.

10. Does Snail1 act to suppress lipogenesis in any other model of fatty liver? Drug induced? MCD diet?

Reviewer #2 (Remarks to the Author):

The study by Liu et al. investigates the insulin-PI 3-kinase-Snail1 pathway linking metabolic signals to epigenetic reprogramming of hepatic de novo lipogenesis. The authors report that insulin up-regulates Snail1 in a PI 3-kinase-dependent manner in both mouse and human hepatocytes and show that this pathway is severely impaired in obesity. Overexpression of Snail1 blocked, whereas deletion of endogenous Snail1 enhanced, insulin-stimulated lipogenesis in hepatocytes. Hepatocyte-specific deletion of Snail1 exacerbated, whereas liver-specific overexpression of Snail1 ameliorated, NAFLD in mice. Ablation of hepatic Snail1 also attenuated NAFLD-associated insulin resistance and glucose intolerance; conversely, liver-specific overexpression had the opposite effect. At the mechanistic level, Snail1 bound to the fatty acid synthase (Fasn) promoter and repressed Fasn promoter activity through histone modifications. This study is of potential interest as it suggests that Snail1 acts as a novel regulator of de novo lipogenesis under physiological and pathophysiological conditions.

Major comments

1-The observation that Snail1 binds to the promoter of Fasn is interesting. Does Snail1 also binds to other lipogenic promoters (Acc, SCD1)? It would be important to extend the role of Snail in the control of lipogenesis by performing large scale analysis of Snail binding (ChIP seq) coupled with a large analysis of Snail target genes (microarrays).

2-What is the mechanism by which Snail act as an epigenetic modifier?

3-Srebp-1c is a well known master regulator of lipogenic gene expression in response to insulin. Is the expression/activity of Srebp-1c modified in the absence or presence of Snail? Is the action of Snail on Fasn independent of Srebp-1c activity (mature form, binding to SRE?).

3-Is Snail expression modified in liver of patients with NAFLD?

Reviewer #3 (Remarks to the Author):

Nonalcoholic fatty liver disease (NAFLD) is a burgeoning health problem and is a major cause of liver

related morbidity and mortality. This manuscript by Liu et al. presents new findings on this important topic. The authors show that insulin upregulates transcriptional repressor Snail, which in turn interacts with HDACs and deacetylates lipogenic gene promoters. Thus Snail appears to put a brake on insulin-stimulated hepatic lipogenesis. Importantly, using both liver-specific loss-of-function and gain-of-function Snail mouse models, they reveal an important role of Snail in amelioration of hepatic steatosis. The experiments were well designed, and the results are novel and very interesting.

Below are the points that may be addressed:

1. Does feeding/fasting regulate Snail expression? Does LXR agonists regulate Snail expression?
2. Fig. 2E, why basal level of lipogenic genes is not affected by overexpression of Snail?
3. Fig. 3, circulating levels of triglycerides and FFA should be provided.
4. Does Snail regulate the expression of LXR and/or SREBP?
5. Fig. 7, data of circulating level of insulin should be provided in KO mice with a chow diet.
6. In addition to high fat diet (Fig. 4), it would be interesting to see whether overexpression of Snail can protect against NAFLD in other mouse obese models.

Response to Reviewer 1

General Comments:

“It is also disappointing that the intro does not provide any information about Snail. To quote from a review written by Kaufhold, “Snail1-induced EMT involves the loss of E-cadherin and claudins with concomitant upregulation of vimentin and fibronectin, among other biomarkers.” It seems important to also measure these markers in the models proposed here”.

Following these suggestions, we expanded introduction by adding more information about Snail1. We also measured all of the mentioned genes and added the new data in the revised Supplementary Fig. 4b. In hepatocytes, Snail1 appears not to regulate expression of E-cadherin (Ecad), claudin1 (Cldh1), vimentin, and fibronectin (Fn1).

“Also it is known that Snail is regulated by PI3Kinase in other models, which does not seem to be noted here”.

We cited an additional PI 3-kinase-related reference in revision: “In line with these findings, Akt2 was reported to mediate TGFβ1 upregulation of Snail1²⁰”.

“With regard to describing fatty liver, the Nile red staining is basically uninformative. It is not clear how many liver sections were examined to describe fatty liver and who examined them. Ideally this should be done by a hepatopathologist who is blinded to the nature of the samples. Thus inflammation and fibrosis can also be scored. Fatty liver can be quite heterogeneous so representative slices can be deceiving”.

Representative slices alone may be, as Reviewer 1 described, less informative. Because of this concern, we measured multiple related parameters (e.g. H&E staining, Nile red staining, liver TAG levels, lipid metabolic pathways) to test the liver steatosis phenotypes. Notably, Nile red and H&E staining of liver sections provided important information about lipid droplet size and number. We clarified that liver sections were coded. Data were collected blindly and separately analyzed. We assessed liver inflammation and fibrosis as suggested, and provided new results in the revised Supplementary Fig. 4b. Snail1 overexpression did not affect expression of proinflammatory cytokines and profibrotic genes.

“While inclusion of data from both ex-vivo and in-vivo models is helpful the switching back and forth within single figures makes the data appear less rather than more coherent. Also it’s never entirely clear why transformed hepatocytes are used”.

We appreciate these comments. However, we received the opposite opinion from other reviewers in several papers. They suggest that describing one pathway/program in depth at the molecular, cellular, and animal levels (in vitro, ex-vivo, and in-vivo) to emphasize the underlying physiology and significance, and then moving to next pathway/program.

HepG2 (human hepatocytes) were used to try to extend the study to human physiology. Ideally, primary human hepatocytes would provide stronger evidence. Unfortunately, we do not have an access to primary human hepatocytes. Additionally, a prolonged cell culture has been known to induce dedifferentiation of primary hepatocytes. In contrast, HepG2 cells are able to maintain their properties in culture.

Specific Points:

1. *“The authors conflate NAFLD with NAFL. Fatty liver as in NAFL, which may or may not be benign may or may not progress to disease which includes NASH, fibrosis, cirrhosis and potentially development of HCC. In this report only fatty liver has been examined”.*

We appreciate these comments. Definition of NAFLD vs NAFL (or NAFLD vs NASH, to a less extent) is confusing in literature. In the current models, liver steatosis is associated with impaired insulin action and glucose metabolism, prompting us to use NAFLD.

2. *“The authors refer to one of potentially 2 explanation for fatty liver in the context of insulin resistance which is that paradoxically hepatic insulin resistance is associated with increase lipogenesis which also implies that it is fatty liver that causes insulin resistance. Several recent reviews point out that there is likely differential sensitivity of liver in comparison to fat with regard to insulin action and that as fat becomes resistant serum insulin rises while the liver is still sensitive which is why fatty liver occurs. This should be included in the narrative”.*

We expanded discussion about this possibility in revision: “...obesity is associated with hyperinsulinemia to compensate for insulin resistance. Particularly, adipose insulin resistance induces hyperinsulinemia when liver is still sensitive to insulin; hyperinsulinemia drives hepatic lipogenesis, further exacerbating NAFLD”.

3. *“Given the potential importance of the association between Snail1 and insulin action, it would be important to have more than a single data point on this interaction. All that is shown is that after six weeks of a high fat diet, when mice are fasted overnight, Snail1 expression is induced by insulin at two hours only in mice fed chow. What happens in the fed state? Were dose response curves performed? If more insulin is given to HFD group will expression be induced”?*

We conducted the requested experiments and added new results in the revised Supplementary Figs. 1e (fed state) and 1g and 1i (dose responses). Liver Snail1 levels are higher in the fed state relative to the fasted state. Insulin resistance decreases the ability of insulin to upregulate Snail1 at all doses tested.

4. *“The HFD is not an HFD – as per the methods the 60% fat diet administered also contains increased amounts of sucrose compared to chow. This should be noted”.*

We clarified the confusion in revision.

5. *“Can doses of insulin and glucose be included in the methods section referring to GTT and ITT”?*

Different doses of insulin (for ITT) and glucose (for GTT) were used to match different mouse metabolic states. For clarity, we described dose information in figure legends.

6. *“Based on the data shown Snail1 does not suppress lipogenesis, it limits insulin induced lipogenesis. This should be clarified. Does Snail1 deletion shift the dose response”?*

We performed new experiments to address these issues. Overexpression of Snail1 directly suppressed baseline lipogenesis (the revised Supplementary Fig. 6c) as well as baseline expression of *Fasn* in primary hepatocytes (the revised Supplementary Fig. 6d-e). These data support the conclusion that Snail1 is able to suppress lipogenesis independently of insulin. Insulin upregulates Snail1 that mediates suppression of lipogenesis. New data showed that Snail1 deficiency causes an upward shift of insulin dose response curves (i.e. increased insulin response) (the revised Supplementary Fig. 2b).

7. *“To reach the conclusion in Figure 4, that overexpression of Snail1 suppresses lipogenesis, livers of mice with AAV mediated increased Snail1 expression are shown in animals fed a high fed diet. The data*

provided are a single H & E slice and overall liver triglycerides. This is not enough to show suppression of lipogenesis which ideally should be looking at fatty acid synthesis and expression of lipogenic enzymes in the liver. Other explanation for decreased liver fat are increased fatty acid oxidation or increased export”.

Expression of hepatic lipogenic enzymes, requested by Reviewer 1, was described in the revised Fig. 5. To further address this concern, we measured expression of the genes that control fatty acid β oxidation (*CPT1 α* and *LCAD*) and VLDL secretion (*MTTP* and *apoB*). Snail1 overexpression did not alter expression of these genes (the revised Supplementary Fig. 4b).

8. “The conclusion appears to be inferred from Figure 5 where deletion of hepatic Snail leads to increased expression of lipogenic enzymes and which is described as Snail directly suppressing liver lipogenesis. However what is shown in figure 5 is enhanced expression of lipogenic enzymes in the absence of snail which is different from showing suppression”.

We clarified a confusion in revision. Liver-specific overexpression of Snail1 decreased both expression of lipogenic enzymes (the revised Fig. 5e-f) and liver steatosis (the revised Fig. 4c-f). In primary hepatocyte cultures, overexpression of Snail1 similarly suppressed *Fasn* expression and lipogenesis (the revised Supplementary Fig. 6c-e).

9. “To be consistent similar data is needed from the model described in Figures 4 and 5”.

We described the liver steatosis phenotypes in the revised Fig. 3 (Snail1-null) and Fig. 4 (Snail1-overexpressing) and the underlying lipogenic pathways in the revised Fig. 5a-d (Snail1-null) and Fig. 5e-f (Snail-overexpressing).

10. “Does Snail1 act to suppress lipogenesis in any other model of fatty liver? Drug induced? MCD diet?”

We tested three additional models (*ob/ob* mice and fructose-induced or MCD-induced liver steatosis) as suggested. MCD-fed mice lost body weight and were sick, so we did not use this model. In *ob/ob* mice, liver-specific overexpression of Snail1 similarly attenuated liver steatosis (the revised Fig. 4e-f). In contrast, deletion of hepatic *Snail1* enhanced fructose-induced liver steatosis (the revised Fig. 3g-h).

Response to Reviewer 2

1-“The observation that Snail1 binds to the promoter of *Fasn* is interesting. Does Snail1 also binds to other lipogenic promoters (*Acc*, *SCD1*)? It would be important to extend the role of Snail in the control of lipogenesis by performing large scale analysis of Snail binding (ChIP seq) coupled with a large analysis of Snail target genes (microarrays)”.

We performed the requested experiments by assaying binding of Snail1 to the *Acc1*, *Acl*, and *SCD1* promoters (within -900 bp from the transcription start site) using ChIP. We did not detect binding of Snail1 to these regions (Fig. for Rev 2). It is likely that Snail1 may bind to other regulatory regions beyond the loci we examined. We performed the microarrays experiments to identify potential Snail1 target genes (the Revised Supplementary Fig. 4a).

Fig. for Reviewer 2. Snail1 does not bind to the *Acc1*, *Acl*, and *SCD1* promoters. Male mice were transduced with Snail1 adenoviral vectors for 3 wks. Occupancy of Snail1 on *Acc1*, *Acl*, and *SCD1* promoters were assessed in the liver by ChIP assays.

We appreciate the suggested CHIP-seq experiments to profile Snail1 occupancies in the entire genome. The studies likely lead to identification of additional Slug target genes. However, unbiased CHIP-seq studies/data analyses are time-consuming, and we are unable to complete them in this work. However, we will perform the experiments in the future.

2-“What is the mechanism by which Snail act as an epigenetic modifier”?

We proposed that Snail1 recruits, via its SNAG domain, HDAC1/2 to lipogenic gene (e.g. *Fasn*) promoters where HDAC1/2 catalyze repressive deacetylation of H3K9 and H3K27.

We performed additional experiments to further test this hypothesis following this comment. To test if pharmacological inhibition of HDAC1/2 abolishes the ability of Snail1 to suppress *Fasn* expression and lipogenesis, we treated hepatocytes with HDAC1/2 inhibitor TSA. TSA completely abrogated the ability of Snail1 to suppress *Fasn* expression and lipogenesis (the revised Supplementary Fig. 6c-e). To test if deleting the SNAG domain (Δ N20) abolishes the ability of Snail1 to suppress lipogenesis, we transduced hepatocytes with Δ N20 adenoviral vectors. Δ N20, unlike Snail1, was unable to suppress the ability of insulin to stimulate lipogenesis (the revised Fig. 6f) and expression of *Fasn* (the revised Supplementary Fig. 6f), and was unable to induce deacetylation of both H3K9 (the revised Supplementary Fig. 6g) and H3K27 (the revised Fig. 6g).

3-“*Srebp-1c* is a well known master regulator of lipogenic gene expression in response to insulin. Is the expression/activity of *Srebp-1c* modified in the absence or presence of Snail? Is the action of Snail on *Fasn* independent of *Srebp-1c* activity (mature form, binding to SRE?)”.

We carried out additional experiments, following these comments. Overexpression of Snail1 increased, whereas deletion of *Snail1* decreased, *Srebp-1c* levels in hepatocytes (the revised Supplementary Fig. 5a-b). Snail1 suppressed the mutant *Fasn* promoter lacking the *Srebp-1c* response element (the revised Supplementary Fig. 5c-d), suggesting that Snail1 represses the *Fasn* promoter independently of *Srebp-1c* (the revised Supplementary Fig. 5d). Snail1 counteracted *Srebp-1c* action (the revised Supplementary Fig. 5e).

4-“Is Snail expression modified in liver of patients with NAFLD”?

It is an important question. Unfortunately, we cannot address this issue, because we currently do not have an access to human samples. We will answer this question through collaboration with clinical investigators in the future.

Response to Reviewer 3

1. “Does feeding/fasting regulate Snail expression? Does LXR agonists regulate Snail expression”?

We conducted the requested experiments. Overnight fasting downregulated liver Snail1 levels (the revised Supplementary Fig. 1e). LXR agonist did not affect Snail1 levels in HepG2 hepatocytes (Fig. a for Rev 3).

2. “Fig. 2E, why basal level of lipogenic genes is not affected by overexpression of Snail”?

We appreciate this question. To measure insulin-stimulated lipogenesis, hepatocytes were cultured in medium containing low glucose (5 mM)/no FBS for over 14 h to reduce baseline expression of lipogenic enzymes. Low baseline expression likely masks detection of additional inhibition by Snail1 under these conditions.

3. “Fig. 3, circulating levels of triglycerides and FFA should be provided”.

We provided the requested data in the revised Supplementary Figs. 3b, 3d, and 3i-j.

4. “Does Snail regulate the expression of LXR and/or SREBP”?

We conducted the requested experiments. Snail1 suppressed Srebp-1c expression (the revised Supplementary Fig. 5a-b), but it did not affect LXR expression (Fig. b for Rev 3).

5. “Fig. 7, data of circulating level of insulin should be provided in KO mice with a chow diet”.

We measured plasma insulin levels in chow-fed mice (*Snail1*^{Δhep}: 0.70 ± 0.07 ng/ml, n=5; *Snail1*^{flox/flox}: 0.61 ± 0.09 ng/ml, n=5; p=0.42), and added the new data in revision.

6. “In addition to high fat diet (Fig. 4), it would be interesting to see whether overexpression of Snail can protect against NAFLD in other mouse obese models”.

We performed the suggested experiments in *ob/ob* mice with genetic obesity. Liver-specific overexpression of Snail1 similarly suppressed liver steatosis in *ob/ob* mice as in high fat diet-fed mice (the revised Fig. 4e-f).

Fig. for Reviewer 3. a. HepG2 cells were treated with LXRα agonist T0901317 for 24 h. Cell extracts were immunoblotted with the indicated antibodies. **b.** C57BL/6 male mice were fed a HFD and transduced with Snail1. Liver was harvested 3 wks after transduction. *Snail1*^{Δhep} and *Snail1*^{flox/flox} mice were fed a HFD for 10 wks. Liver LXRα mRNA levels were measured by qPCR and normalized to 36B4 levels. n=6 per group.

REVIEWERS' COMMENTS:

Reviewer #1 (Remarks to the Author):

This manuscript reports on the interaction of the transcriptional repressor Snail1 to regulate lipogenesis through insulin mediated pathways. The essential data is that insulin treatment increases snail1 expression. Snail 1 deletion leads to increased lipogenesis in mice eating high fat diet while overexpression of snail decreases lipogenesis. This is shown using a number of models including genetic deletion of snail1 using cre-lox in which case there is higher expression of lipogenic enzymes and then AAV overexpression showing reduced expression of lipogenic enzymes. Lipid accumulation is also demonstrated and shown to be consistent with increased lipogenesis without snail and inhibition with snail overexpression. The mechanism for snail1 effects on lipid accumulation is at least in part attributable to epigenetic regulation of the FASN promoter.

Based on these findings the authors conclude that the induction of snail by insulin treatment represents an insulin mediated pathway to suppress lipogenesis. While the data presented are of interest the conclusion that through snail1 insulin can suppresses lipogenesis in the liver isn't to far a reach. Insulin induces multiple changes in the liver and it is very possible that snail is a downstream consequence of multiple other effects of insulin via an unknown compensatory pathway.

Furthermore the data in figure 7 on the effects of snail1 on insulin action are pretty weak. The ITT is marginally worse only at one time point, the GTTs almost completely overlap in mice with snail1 liver specific deletion. The insulin levels are higher, however these mice, based on previous data have fatty liver because of snail deletion. In mice that have increased snail insulin sensitivity is slightly increased based on the fall in glucose on ITT and the GTTs are again pretty much overlapping. Even if the differences are statistically significant they don't seem sufficient to be physiologically relevant, especially if these livers have less lipid accumulation. Also baseline insulin levels are not shown for the mice over-expressing snail1.

So overall the idea that snail1 bifurcates insulin action is a reach. No other factors that affect fatty liver are considered in this context including glucagon, glucocorticoids, FGF21 or beta agonists.

Specific comments: better descriptions of the figure legends are necessary across the board.

NAFLD is very complex and the overview provided by the authors is very narrow and there is a tendency for "self-references" for reviews. There may be significant benefit in broadening the picture by reading and referring to a review by David Moore or Jay Horton.

Reviewer #2 (Remarks to the Author):

no more comments

Reviewer #3 (Remarks to the Author):

The authors have addressed my concerns.

Response to Reviewer #1

“This manuscript reports on the interaction of the transcriptional repressor Snail1 to regulate lipogenesis through insulin mediated pathways. The essential data is that insulin treatment increases snail1 expression. Snail 1 deletion leads to increased lipogenesis in mice eating high fat diet while overexpression of snail decreases lipogenesis This is shown using a number of models including genetic deletion of snail1 using cre-lox in which case there is higher expression of lipogenic enzymes and then AAV overexpression showing reduced expression of lipogenic enzymes. Lipid accumulation is also demonstrated and shown to be consistent with increased lipogenesis without snail and inhibition with snail overexpression. The mechanism for snail1 effects on lipid accumulation is at least in part attributable to epigenetic regulation of the FASN promoter.

Based on these findings the authors conclude that the induction of snail by insulin treatment represents an insulin mediated pathway to suppress lipogenesis. While the data presented are of interest the conclusion that through snail1 insulin can suppresses lipogenesis in the liver isn't to far a reach. Insulin induces multiple changes in the liver and it is very possible that snail is a downstream consequence of multiple other effects of insulin via an unknown compensatory pathway. Furthermore the data in figure 7 on the effects of snail1 on insulin action are pretty weak. The ITT is marginally worse only at one time point, the GTTs almost completely overlap in mice with snail1 liver specific deletion. The insulin levels are higher, however these mice, based on previous data have fatty liver because of snail deletion. In mice that have increased snail insulin sensitivity is slightly increased based on the fall in glucose on ITT and the GTTs are again pretty much overlapping.

Even if the differences are statistically significant they don't seem sufficient to be physiologically relevant, especially if these livers have less lipid accumulation. Also baseline insulin levels are not shown for the mice over-expressing snail1”.

We showed that ablation of endogenous hepatic Snail1 exacerbates HFD-induced hyperinsulinemia, impairs liver insulin signaling, and decreases the ability of insulin to lower blood glucose. These results indicate that endogenous hepatic Snail1 physiologically regulates insulin sensitivity and glucose metabolism. Notably, the effect of hepatic Snail1 on glucose metabolism, unlike its profound effects on liver lipogenesis, is modest.

“So overall the idea that snail1 bifurcates insulin action is a reach. No other factors that affect fatty liver are considered in this context including glucagon, glucocorticoids, FGF21 or beta agonists”.

We appreciate these comments. We will explore, in the future, the possibility that Snail1 may be involved in mediating regulation of lipid metabolism in response to other factors. In the Discussion section, we proposed that “Aside from insulin, Wnts, TGFβ1, and additional factors also upregulate Snail1.... Snail1 may serve as a common node upon which these factors converge to regulate lipogenesis and intracellular lipid content”.

“Specific comments: better descriptions of the figure legends are necessary across the board. NAFLD is very complex and the overview provided by the authors is very narrow and there is a tendency for “self-references” for reviews. There may be significant benefit in broadening the picture by reading and referring to a review by David Moore or Jay Horton”.

We expanded the overview of NAFLD by adding a new paragraph in the revised Discussion section, following these comments. We also made changes in figure legends.

Response to Reviewer #2

"no more comments".

We appreciate Reviewer 2's evaluation of this work.

Response to Reviewer #3

"The authors have addressed my concerns".

We appreciate Reviewer 3's assessments of this work.